# A Miniature Permanent Magnet Assembly with Localized and Uniform Field with an Application to Optical Pumping of Helium

**Garnet Cameron ***, Jonathan Cuevas, Jeffrey Pound, Jr. and David Shiner

Physics Department, College of Science, University of North Texas, 210 Avenue A, Denton, TX 76203, USA; JonathanCuevas@my.unt.edu (J.C.); jeffreyPound@my.unt.edu (J.P.J.); shiner@unt.edu (D.S.)
* Correspondence: garnetCameron@my.unt.edu

**Featured Application: atomic laser spectroscopy.**

**Abstract:** Atomic state preparation can benefit from a compact and uniform magnetic field source. Simulations and experimental measurements have been used to design, build, and test such a source and then apply it to the optical pumping of atomic helium. This source is a 9.5 mm (3/8″) OD × 6.7 mm (1/4″) ID × 9.5 mm (3/8″) long, NdFeB-N42 assembly of 1.6 mm (1/16″) thick customized annular magnets. It has octupole decay with a residual dipole far field from imperfect dipole cancelations. Fast B-field decay localizes the field, minimizing the need for shielding in applications. It has a greater than 50% clear aperture with a uniform and collimated magnetic field consistent with the prediction of several models. The device is applied to a high precision $^{3,4}$He laser spectroscopy experiment using $\sigma^+$ or $\sigma^-$ optical pumping currently resulting in a measured 99.3% preparation efficiency and in accordance with a rate equation model.

**Keywords:** optical pumping; annular permanent magnet; miniature magnet; laser orbital angular momentum; collimated magnetic field; fringe magnetic field





## 1. Introduction

In atomic state preparation, often a circular-polarized, broadband laser uses orbital angular momentum transfer to migrate through multiple transitions to a desired bounding spin state [1]. A magnetic field can be used to define and maintain the quantization z-axis. The physical dimensions of the magnetic field and its source can be a significant concern. Ultra-high vacuum (UHV) experimental system components often vie for cavity space and can force trade-offs in overall experimental optimizations. An intended application of this magnet assembly is an existing atomic laser spectroscopy experiment that has evolved through at least three versions since 1995 [2–4], starting from $^4$He with 51 ppb precision results [2]. It is now capable of simultaneous measurements of $^3$He and $^4$He isotopes at 20 ppb [4].

The atomic beam apparatus is shown in Figure 1. Metastable helium atoms (He*) are prepared into $1s^1 2s^1$ electronic configuration via ~30 eV electron bombardment in the helium source at the right end of the green atomic beam. An approximately 25–75 mixture of singlet and triplet He* atoms is created. The triplet, 2S, $m_j = 0$ ($2^3 S_{1, m_j=0}$) Zeeman level is prepared as the experiment signal channel by optically pumping to zero population. Preparing the atomic beam with a circular-polarized laser instead of the previous linearly polarized laser doubles our signal size by sweeping the atoms into either +1 or −1 magnetic state versus splitting the population between the two. Retarders are used to transform an incoming linear-polarized 1 μm laser to $\sigma^+$ or $\sigma^-$ polarization.

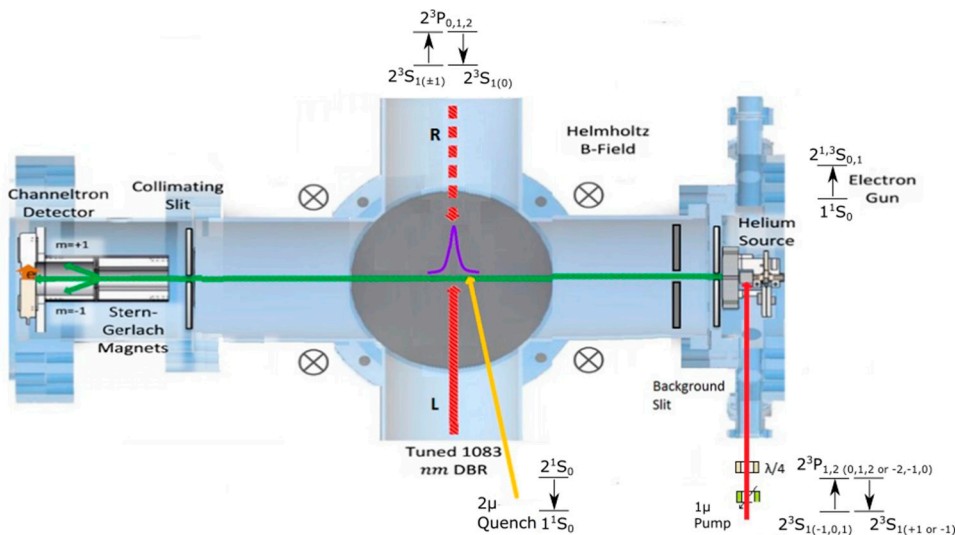

**Figure 1.** Atomic beam apparatus: Helium Source is the area of interest.

According to quantum mechanical selection rules, increasing or decreasing angular momentum $m_j$ by laser transition demands (X)(Y) transitions or $\Delta m_j = \pm 1$ [4]. The laser E-field (polarization) should be perpendicular to the quantization axis (z) defined by an imposed B-field ($\vec{B}_{magnet}$). As suggested by Kastler [1], laser propagation ($\vec{k}$) parallel to $\vec{B}_{magnet}$ guarantees this requirement for a circularly polarized laser, as shown in Figure 2. A permanent magnet is a passive and compact magnetic field source versus a high current and turns-dense solenoids with associated heating issues. Since the 1980's, NdFeB permanent magnets have offered the highest available range of maximum energy products [5]. N42 grade is an economical choice for prototyping but carries a low Curie point of 80 C. We focus on the design, construction, and application of the preparation laser magnet.

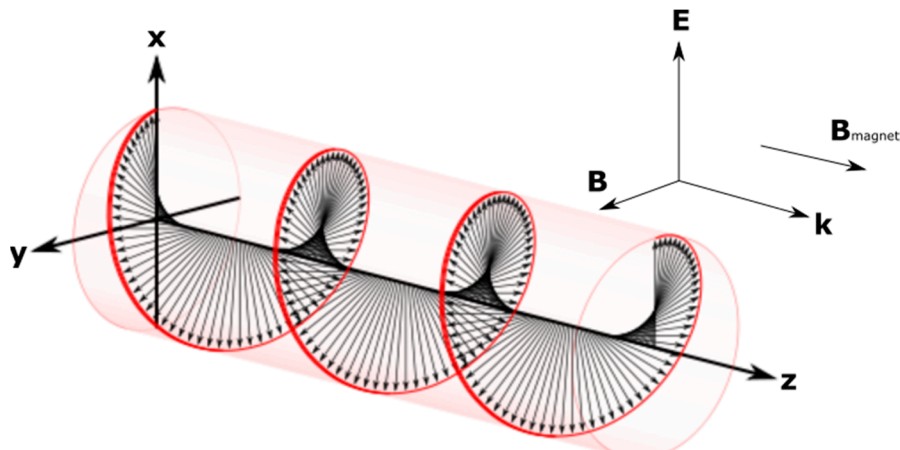

**Figure 2.** Atomic beam preparation laser polarization, **k**, and **B$_{magnet}$** arrangement (Right or σ⁺ shown).

## 2. Magnet Requirements and Dimensions

Our atomic experiment collects $10^4$ counts in each counting interval. The Poisson noise threshold is therefore 1/100 using $\varepsilon = \sigma / \sqrt{N}$, where $\varepsilon$ is error, $\sigma$ is standard deviation, and N is quantity of events [6]. This threshold is expected to continue into projected higher signal levels coupled with reduced count interval time length. Less than 0.1 radian deviation of the magnetic field from the z-direction is required to suppress competing depolarizing transitions to less than 1/100. Gaussmeter measurements around the neighboring electron

gun magnets that use a return flux yoke revealed a 5 G perpendicular surrounding field suggesting a 50 G minimum preparation magnetic field strength.

Any fringing field in the spectroscopic laser interaction region needs to be minimized. In addition, Helium energy levels change at ~1.4 MHz/G, so sub-kHz precision in helium spectroscopy makes field gradients of ~1 mG/mm desirable. The atomic beam pump laser is approximately 3.5″ from the interaction region, as seen in Figure 1. The far field is thus defined as greater than 3″ removed from the preparation magnet center. Shielding would impact other needed magnetic fields in the apparatus.

For compactness and convenience, the assembly should fit in the existing space for linear-polarized atomic beam pumping. The critical dimensions are 3/8″ along the atomic beamline (outer diameter of the annular magnet), and enough atomic beam pumping transit distance to allow a minimum of 90% and preferred 99% or better pumping to $m_s$ = +1 or −1 (internal diameter). In Figure 3, the upper left corner shows a cross-sectional layout.

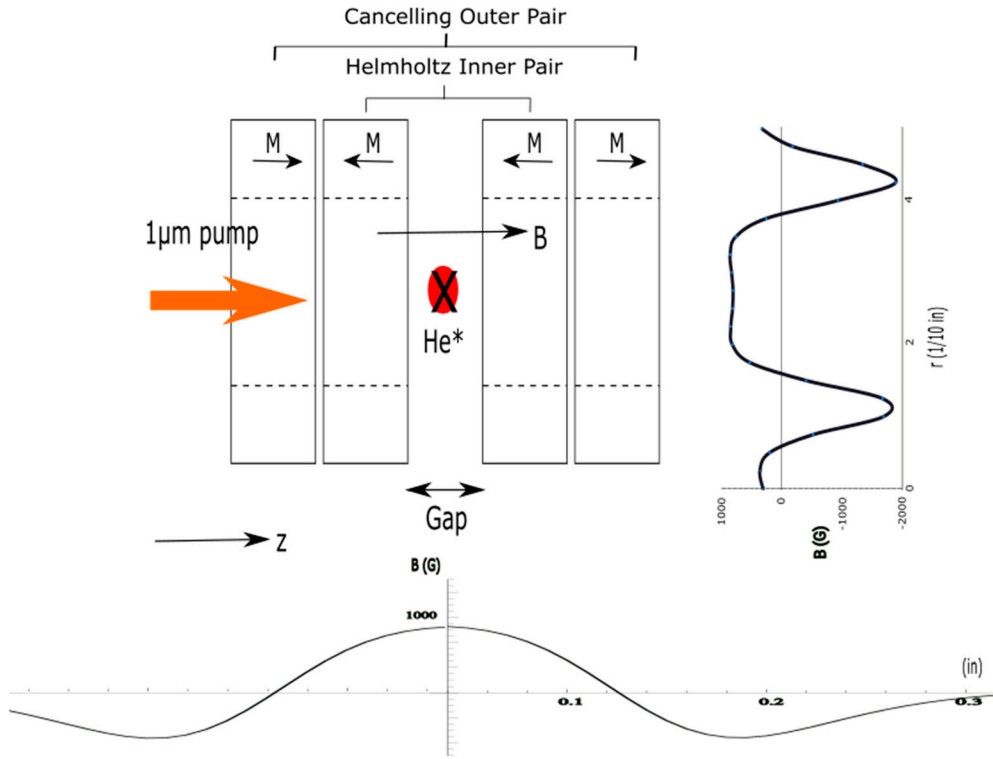

**Figure 3.** Created magnet assembly: bottom, axial (z) $B_z$ field; right: mid-plane transverse (x or $\rho$) $B_z$ field.

Faraday isolators such as Gauthier's design [7] offer some inspiration toward the fringe field requirement. As shown in Figure 3, stacked pairs of magnets with opposed magnetization sum to a zero far field. The central field is collimated in a modified Helmholtz geometry as the inner surfaces of the inner magnet pair make the most contribution to the field in the pumping region (see the current loop model discussed below).

Note that coincident points on axial (z-axis) and transverse (x-axis or $\rho$ due to cylindrical symmetry) B-field profiles within the gap must have the same value, i.e., the physical center point. Axial (z) and transverse (x or $\rho$) $B_z$-field plots are shown in the bottom and top-right of Figure 3, respectively. Greater than three inches from the origin (center of the gap) is the far field. The gap is labeled.

In Section 3.1, we discuss a simple current loop model for each magnet to verify that the physical constraints can support the $B_z$ performance requirements. In Section 3.3, benchtop prototypes, experimental magnetic field measurements, and 3-D simulation models are used to detail and validate various arrangements against our requirements and to allow the selection of an optimum configuration. In Section 4, adequate far-field behavior is verified by Gauss meter measurements and compared to COMSOL modeling.

In Section 5, the central field performance, as well as the overall experimental optical pumping efficiency of atomic helium, is presented. Pumping results are also compared to the predictions of electronic transition rate equations.

## 3. Design and Construction

Magnetic fields are solved in two ways. A quick axial/longitudinal (z-axis) solution can be determined by Biöt-Savart current coil model summation. Alternatively, finite element analysis (FEA) using Maxwell's equations [8] provides a more complete solution profiling longitudinal or z-axis and transverse or x-axis B-fields.

$$
\begin{aligned}
\vec{H} &\equiv -\vec{\nabla}\Phi \\
\vec{\nabla} \cdot \vec{B} &= 0 \\
\vec{B} &= \mu_0\left(\vec{H} + \vec{M}\right)
\end{aligned}
\tag{1}
$$

$H$ is magnetic field strength, $\Phi$ is scalar magnetic potential, $B$ is magnetic field intensity, and $M$ is magnetization. Outside the magnetic material, $M = 0$:

$$
\vec{\nabla} \cdot \vec{B} = \mu_0\left(\vec{\nabla} \cdot \vec{H} + \vec{\nabla} \cdot \vec{M}\right) \quad \Rightarrow 0 = \mu_0\left(\vec{\nabla} \cdot \left(-\vec{\nabla}\Phi\right) + 0\right) \quad \Rightarrow 0 = \nabla^2\Phi
\tag{2}
$$

Inside magnetic material bulk of constant magnetization, $\vec{M}=$ constant. Magnetization may change on the magnet edges due to mutual coercion with other magnets. This effect is a small contribution as indicated by close agreement of measured and COMSOL-simulated [8] values of the B-field (detailed in Section 4). The COMSOL model did not include mutual coercion.

$$
\vec{\nabla} \cdot \vec{B} = \mu_0\left(\vec{\nabla} \cdot \vec{H} + \vec{\nabla} \cdot \vec{M}\right) \quad \Rightarrow 0 = \mu_0\left(\vec{\nabla} \cdot \left(-\vec{\nabla}\Phi\right) + 0\right) \quad \Rightarrow 0 = \nabla^2\Phi
\tag{3}
$$

For FEA modeling, $\vec{B}$ parallel to the surface or *Dirichlet* boundary condition (BC) is applied to the far bounding surfaces while $\vec{B}$ perpendicular to the surface or *Neuman* BC is employed at symmetry faces involving or very close to the magnet(s), e.g., the floor in Figure 4. The model space is reduced by exploiting symmetry planes to minimize simulation overhead. The whole solution is generated by duplicating the minimized volume.

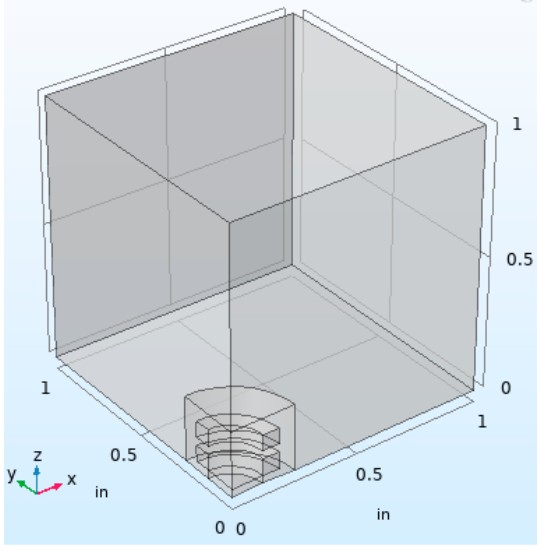

**Figure 4.** Reduced FEA simulation volume.

Exploration FEA simulations shown in Figure 5 examine different configurations of similar annular magnets. Clearly, a parallel magnetized pair stabilizes the central field. The addition of a repelling pair (one on each end) reduces the far-field decay distance.

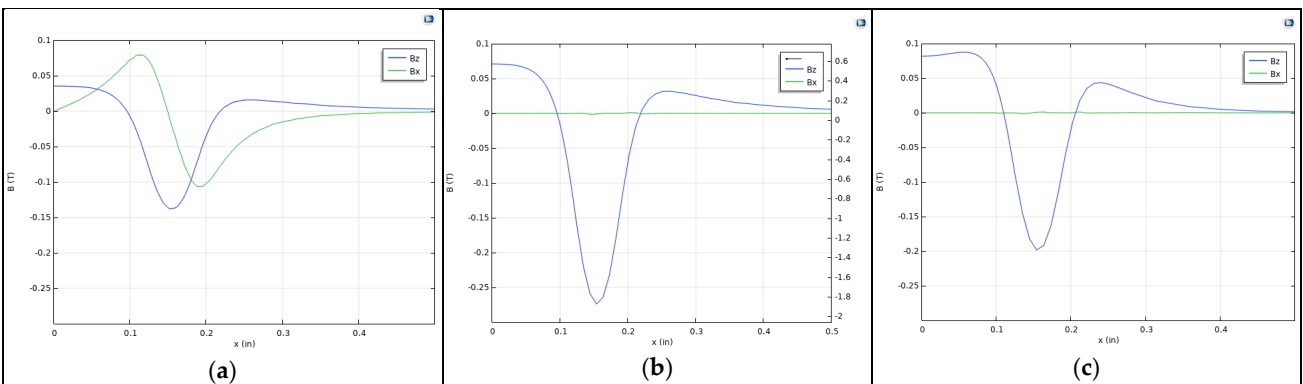

**Figure 5.** Simulated transverse $B_z$ and $B_x$ on z = 0 plane for: (**a**) single; (**b**) inner pair only; and (**c**) full stack (no space between inner and outer pairs).

### 3.1. B-Field Collimation

The required magnet dimensions were initially studied by Biöt-Savart current coil approximation simulations [9], exploring axial B-field strength along the magnet assembly z-axis as the gap size is varied, as seen in Figure 6a. Code for all models are provided at https://github.com/garnetc/CircularLaserMagnet (accessed on 16 September 2021). Each magnet is modeled as a pair of inner and outer coaxial, z-centered, and counter-rotating current loops. Currents more uniformly distributed over the magnet surfaces were simulated and found unnecessary for this initial analysis. The summation of infinitesimal magnetization current loops across the magnet bulk yields the current loops model [9].

From the Biöt-Savart law:

$$d\vec{B} = kI\frac{\left(d\vec{l} \times \vec{r}\right)}{|\vec{r}|^3} \tag{4}$$

where $k$ is the magnetic constant, $I$ is the current, and $r$ is the point of interest position. The well-known analytical solution of the axial field for a circular coil is [10]:

$$B_{z,axial} = \frac{\mu_0 I}{2}\frac{r^2}{\left(z^2 + r^2\right)^{\frac{3}{2}}} \tag{5}$$

Therefore, the assembly axial field $B_z(z)$ is a summation over all the coils/magnets:

$$B_z(z) = \frac{\mu_0 I}{2}\left\{\sum_{\substack{i = 2,3 \\ j = 1,2}} \frac{(-1)^j r_j^2}{[(z - z_i)^2 + r_j^2]^{\frac{3}{2}}} - \sum_{\substack{i = 1,4 \\ j = in, out}} \frac{(-1)^j r_j^2}{[(z - z_i)^2 + r_j^2]^{\frac{3}{2}}}\right\} \tag{6}$$

The magnets are numbered left to right, 1 to 4, and are indexed as $i$. The inner and outer coils are represented by $j$ and are numbered 2 and 1, respectively.

Antiparallel magnetization directions sum depending on gap size. Axial (z-axis) field strength is also influenced by the ratio of inner radius ($R_i$), outer radius ($R_o$), and thickness of each magnet. In Figure 6c, solenoid simulation identifies the optimum thickness and $R_i/R_o$ values. However, thickness and outer diameter (OD) selection were constrained by commercially available options: 1/8″ or 1/16″ and OD ($2R_o$) desired to 3/8″ (mentioned

earlier), respectively. Conceptually, a smaller thickness would be preferable when the annular magnet is modeled as inner and outer solenoids. Thicker magnets relative to outer diameter tend to the infinite coil solution where the axial fields sum vanishes. Axial fields of the inner and outer coils are equal and opposite. Jackson's [9] (p. 225) solenoid-faces-subtending-angles solution shows this behavior, see Figure 7.

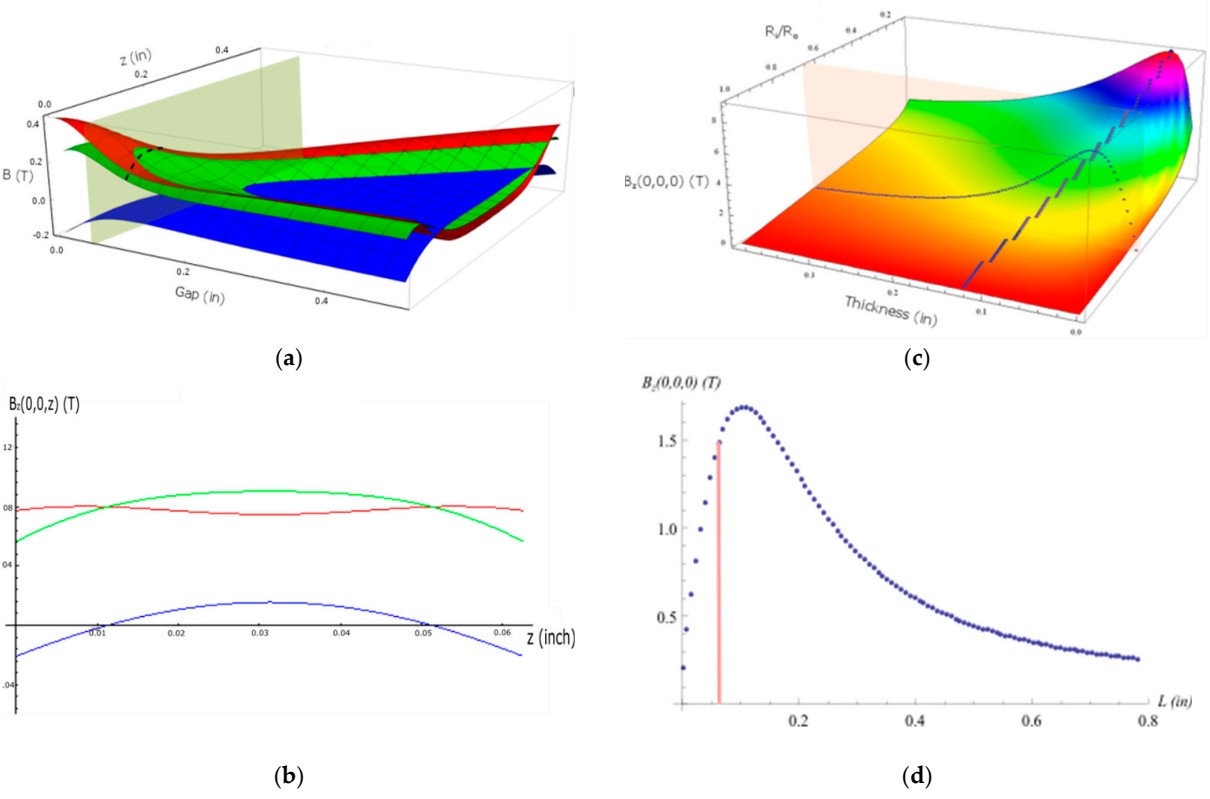

(a)

(c)

(b)

(d)

**Figure 6.** Simple Biöt-Savart axial (z-axis) simulation plots: (**a**) $B_z$(z, Gap)—attracting pair (red), repelling pair (blue), sum (green), system position slice and sum curve (shaded plane and dotted line); (**b**) $B_z$(z, Gap = 0.125″), 0.125″ Gap size due to coils modeled at the center point of 1/16″ thick magnet, i.e., actual gap (1/16″) + 2 * coil offset (1/32″)—attracting pair (red), repelling pair (blue), sum (green); (**c**) single magnet face center $B_z$(Ri/Ro, thickness)—colored Ri/Ro iso-bands highlight ratio behavior, maximum values (black dots), system slice and curve (shaded plane and green broken curve); (**d**) single magnet face center $B_z$(for Ri/Ro = 0.67) vs. thickness L—system position at red line.

$$B_{z,axis}\left(\theta_1,\theta_2\right)=\frac{\mu_0 NI}{2}\left(\cos\theta_1+\cos\theta_2\right)$$

$$B_{z,axis}\left(\theta_1,\theta_2\right)=\lim_{\theta_1=\theta_2\to 0}\mu_0 NI$$

**Figure 7.** Thick-magnet axial field behavior (N = coil turns per length, I = current).

In other words, the outer loop's negative impact on the sum axial ($B_z$) field is reduced by the largest possible OD. Experiment apparatus 3/8″ available space, therefore, sets the OD.3/8″ OD constraint and commercial unavailability of 3/8″ × 1/4″ × 1/16″ forced customization of 3/8″ × 1/8″ × 1/16″ via Dremel-lathe working to widen the 1/8″ inner diameter (ID) to 1/4″ to allow sufficient atom transit time as discussed shortly. Commercial annular magnets are sintered powders encased in a thin metallic jacket; in this case, Ni-Cu-Ni. Usual cutting tools result in fracturing and disintegration. Grinding proved to be a viable option. The four customized annular magnets were evaluated for geometric individuality. A total of 0.7% variance in thickness was the most differing characteristic noted. Figure 8a shows the typical measured transverse (x-axis) $B_z$ profile for a single annular magnet. The center face "surface" field is ~400 G, and the center ring surface field

($B_{ring}$) is ~1600 G. K&J Magnetics (manufacturer) [11] details the original 1/8″ ID annular $B_{ring}$ to be ~1600 G. However, more precise face center measures exhibited ~10% departures from theoretical/simulated values by K&J Magnetics for the 1/4″ ID, i.e., 441 G versus 487 G. Our precise measurements took account of a Lakeshore 450 Gauss meter 30 mils (1 mil = 0.001″) standoff of the active area from the contacted surface. Some hypotheses to explain the reduced field include localized ID grain boundary damage due to grinding or overall magnetization reduction during grinding despite cooling precautions. As a result, all in-house simulation work is referenced to the gap center measured B-field.

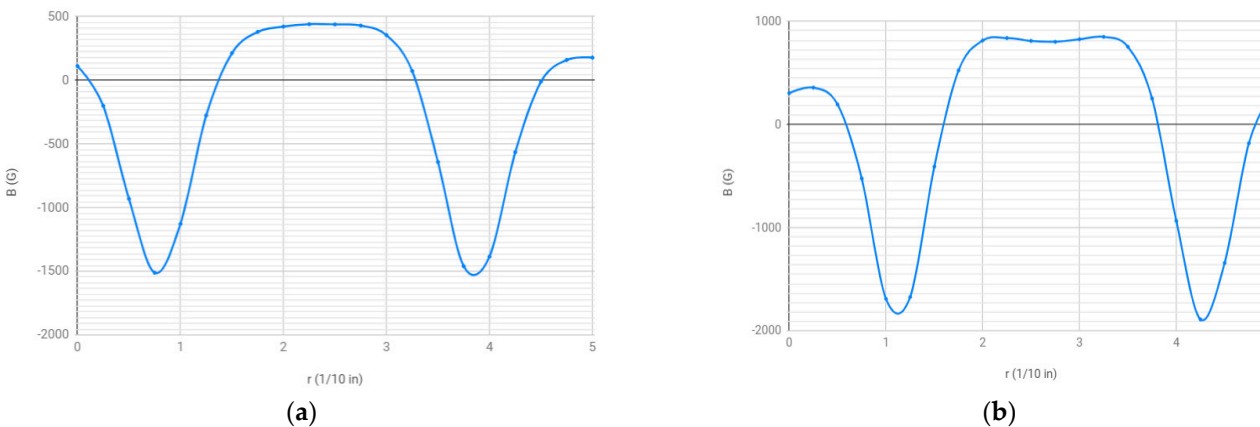

(**a**)          (**b**)

**Figure 8.** Magnet transverse B-field profiles (3/8″ × 1/4″ × 1/16″): (**a**) single; (**b**) assembly.

Colloidal graphite was used to stabilize the exposed surfaces and minimize stray pump laser reflections. An ultra-high vacuum (UHV) chamber compatible mount was manufactured retaining the 1/16″ gap, illustrated in Figure 9. A total of 80 and 180 mil separation between the top and bottom brackets versions allowed vacuum experiment and bench probing, respectively. The transverse Gauss meter wand is ~160 mil wide. Profile results are shown in Figure 8, revealing a 0.1″ to 0.15″ collimated B-field diameter in (b).

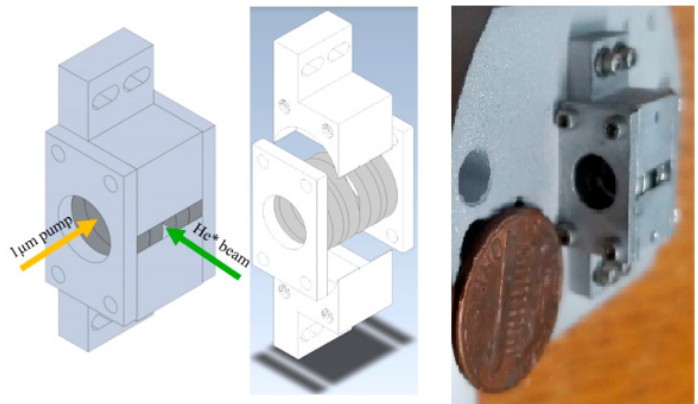

**Figure 9.** UHV optical pumping magnet mount.

1/4″ ID was selected based on minimum atom travel distance, d, dictated by the number of atomic beam cycles, n, to allow 1/100 depletion of the $2^3S_1$:$m_j$ = 0 level. The pump/applied laser will be broad bandwidth covering all relevant 2S-2P transitions. A simplistic estimate of the signal channel population, N, is achieved from the single excitation remaining ratio (portion of atoms still in $2^3S_1$:$m_j$ = 0 state), $r$, excitation lifetime, $\tau$, number of excitations/cycles, $n$, and atom velocity, $v_{rms}$, in the following relations:

$$N = N_0 r^n \Rightarrow n = \frac{log\left(\frac{N}{N_0}\right)}{log r} \tag{7}$$

$$d = v_{rms}(n\tau) = \frac{v_{rms}\tau log\left(\frac{N}{N_0}\right)}{logr} \tag{8}$$

A simplified three-level transition scheme [12] is shown in Figure 10. Atoms are coherently excited from Level 0 (representing our $2^3S_1:m_j = 0$) to Level 2 (all excited states). Relaxation de-coherently branches equally between Level 0 and Level 1 (destination spin state). Therefore, r is 0.5, and τ is treated as twice the natural lifetime due to Level 0–Level 2 cycling.

$$v_{rms} \cong 1500 \text{ m/s} \tag{9}$$

$$\tau = 2 \times 100[\text{ns}] = 200 \text{ ns} \tag{10}$$

$$\frac{N}{N_0} = 0.01 \tag{11}$$

$$r = 0.5 \tag{12}$$

$$\therefore d = 1500 \times 2E - 7 \times log(0.01)/log(0.5) = 1.99 \text{ mm} = 0.078 \text{ in} \tag{13}$$

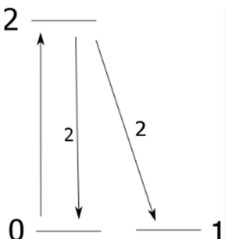

**Figure 10.** Simplified three-level schematic (normalized branching ratios labeled on arcs).

This minimum of 0.08 inch He* beam travel distance requirement is an infinite power result for an average velocity. In addition, alignment tolerances and edge inhomogeneity may also degrade performance, so we take a 0.15-inch diameter of collimated B-field as a more realistic minimum estimate.

### 3.2. Atomic Beam B-Field Collimation Precision

Note that field profile shapes in Figures 7b and 11a are along the z-axis, while the experiment data vs. model in Figure 11b are fixed at z = 0 and along the perpendicular direction (x or y).

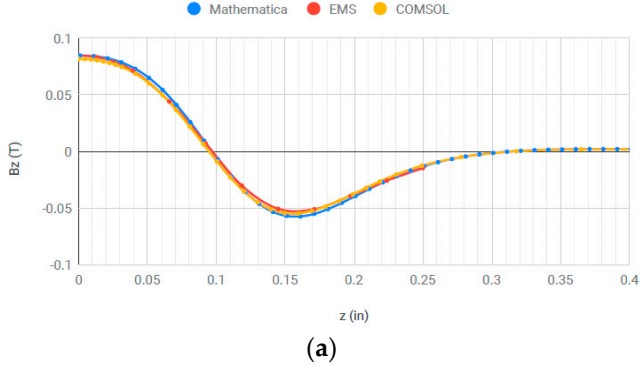

(**a**)

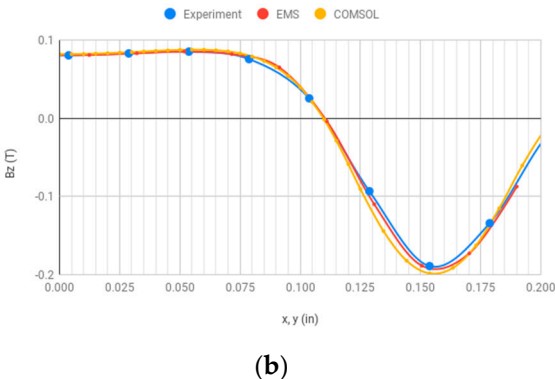

(**b**)

**Figure 11.** $B_z$ Combined plots—origin at symmetry axis: (**a**) longitudinal (z-axis); (**b**) transverse (x-axis). Experiment values are measured with a transverse Gauss meter mounted to a micrometer-controlled translation stage.

The atomic beam moves in the x-direction and has a rectangular cross-section. It is 2 mils (50 μm) wide in the magnet longitudinal direction (z) and 80 mils (2 mm) tall (y). Longitudinal (z) B-field variation has a small effect across the 2 mil range. However,

transverse (x) B-field variation has a significant impact across the pump laser beam diameter. The empirically measured dip shown in Figure 8b is 50 G compared to 800 G at the shoulders (~6%). The effect of the non-zero Bx over the laser pumping region is considered in more detail below (Section 5).

### 3.3. 3-D Simulations

We use 3-D simulations to model more completely the physical situation (still assuming z-axis rotational symmetry) therefore providing more accurate values and both axial (z-axis) and transverse (x-axis) profiles. EMS [13], a SolidWorks add-in by EM Works, and COMSOL (along with the Biö-Savart summations and experimental readings) were used to produce the longitudinal (z-axis) and transverse plots (x-axis) shown in Figure 11.

The simple axial/longitudinal Biö-Savart Mathematica-based discrete current loop model compares favorably with EMS and COMSOL simulations. Similarly, EMS and COMSOL fit well with transverse experimental measurements in most of the gap region. However, some separation is noted near the ID edge (0.125″) and across the annular rims that end at 0.1875″. EMS tracks closer. EMS operates on a hysteresis curve model while COMSOL was pinned to constant magnetization in the magnet bulk. One may hypothesize that some small mutual coercive effects are present in the annular rim. In addition, some aperture effect occurred for the Lakeshore 40 mil diameter active area probe. The 50 G central dip is apparent in the transverse plot (x-axis).

## 4. Far-Field Performance

Minimizing the stray field, ideally below our measurement uncertainty of 0.01 G, at the 3.5″ distant interaction region is another important requirement for this magnet. Helium energy levels change at ~1.4 MHz/G. Measuring frequencies to kHz precision benefits from mG B-field uniformity. Significant fringe field differences are noted for small spacing changes in Helmholtz inner pair and canceling outer pair, as shown in Figure 12. Greater attention is paid to the z = 0 far-field behavior because the subsequent precision spectroscopy interaction region is largely in the z = 0 plane of the magnet assembly (see Figure 1).

(At 3″: 0, 2/32 = −0.020 G; 0, 4/32 = 0.038 G; 1/32, 2/32 = 0.066 G; 2/32, 2/32 = 0.047 G)

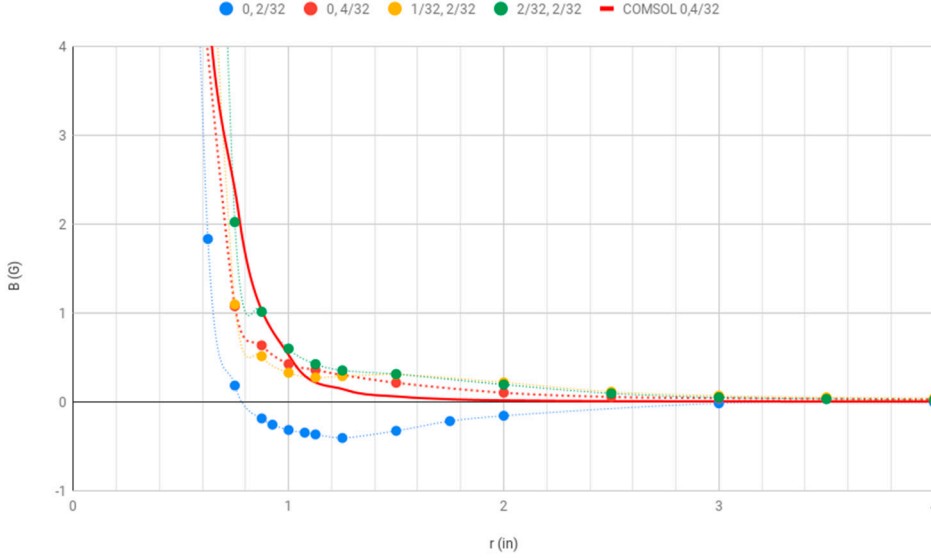

**Figure 12.** Transverse ($\rho$) far-field $B_z$ measurements for variations in the inner-outer pair separation (first value) and gap (second value).

The 1/16″ inner pair gap with no inner to outer spacing (0, 2/32) configuration exhibits anomalous behavior with a negative B-field inflection point. It nevertheless meets the 3″ far-field requirement with better performance than all other configurations presented.

Transverse (x) asymptotic behavior was explored using the (0, 4/32) configuration to allow a more detailed model comparison, as shown in Figure 13. Bz-field roll-off is compared with $1/r^5$ decay (blue broken line) and the COMSOL model prediction. At sufficiently large distances (close to the noise level of our measurements), it appears that the imbalance in the individual canceling dipoles of the configurations leads to a residual dipole component in the decay. Our data and fits indicate that the inner and outer dipole pairs have an approximate 10% mismatch in magnitude. Careful matching should be implemented if a smaller fringe field is required. However, the far-field gradient at 3.5″ is an acceptable 0.5 mG/mm.

$$\frac{dB_z}{dx}(3.5'') = 13\ \text{mG/in} = 0.51\ \text{mG/mm} \tag{14}$$

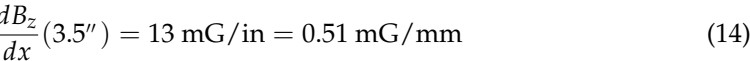

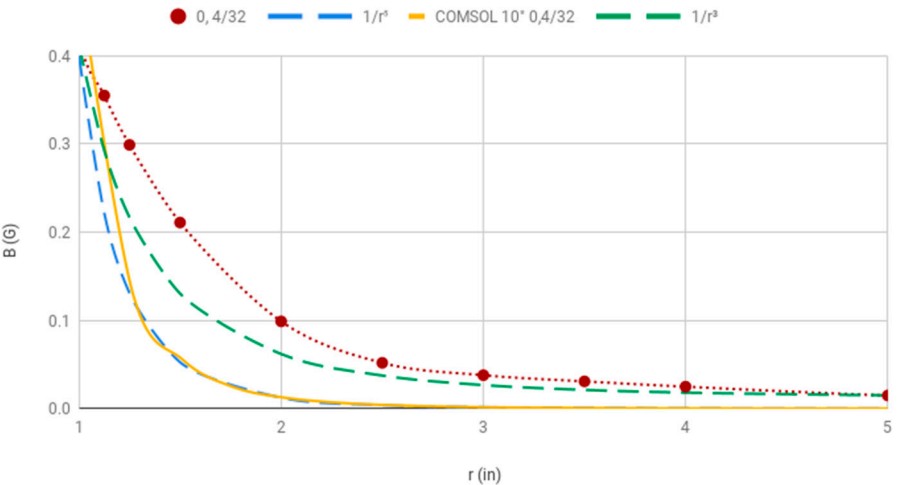

**Figure 13.** Far-field asymptotic behavior, $1/r^5$ (blue broken line), and $1/r^3$ (green broken line).

## 5. Collimation Performance

Sensitivity to z-axis atomic beam offset was investigated via simulations. Findings are shown in Figure 14. Atomic beam z-position offsets as large as 30 mils from z = 0 can be tolerated for x < 0.08″, based on our 1/10 or 80 G $B_x$-threshold requirement as seen in (b). Better than 0.005″ (5 mils or 0.13 mm) of positioning precision is typical for the apparatus. However, for x > 0.08″, even a 1 mil z-offset (purple curve) shows a significant increase, reaching ~50 G at x = 0.125 (ID edge). In practice, exposure to $B_x$ greater than 80 G is likely beyond x = 80 mils. Polarization purity would be adversely affected by this field on the atomic beam exit side of the preparation area.

The figure of merit for He* preparation is the survival fraction of $2^3S_{1,\ m_j=0}$ remaining from the He* mixture produced in the electron-bombardment source. The relevant radiative excitation and relaxation paths for $\sigma^+$ ($\Delta m_s = 1$) are depicted in Figure 15. $\sigma^+$ and $\sigma^-$ excitations are symmetric about $m_s = 0$.

The following rate equation quantifies the population behavior at level i, $N_i$, based on normalized electric dipole matrix elements $D_{ik}$ for transitions between levels i and k as dictated by the laser polarization, $\sigma$, e.g., $\sigma^+$ polarization [14,15] (p. 518). Equal laser power is assumed across all energy levels.

$$\frac{dN_i}{dt} = \sum_{k=\{\sigma\}} D_{ik}N_k - D_{ik}N_i \tag{15}$$

Figure 16 shows the survival ratio with increasing laser power. The preparation laser is verified to >1/1000 circular polarization purity by the crossed-polarizer method. Two features are apparent in (a): a high clearing and our required optical pumping efficiency are achieved, but the model predicts sharper population decline than observed. The

best ratio is ~99.3%, with the other extreme spin state below the noise level of ~0.1%. The Bz fringe effect presented in Figure 14b is theorized as the cause for higher survival.

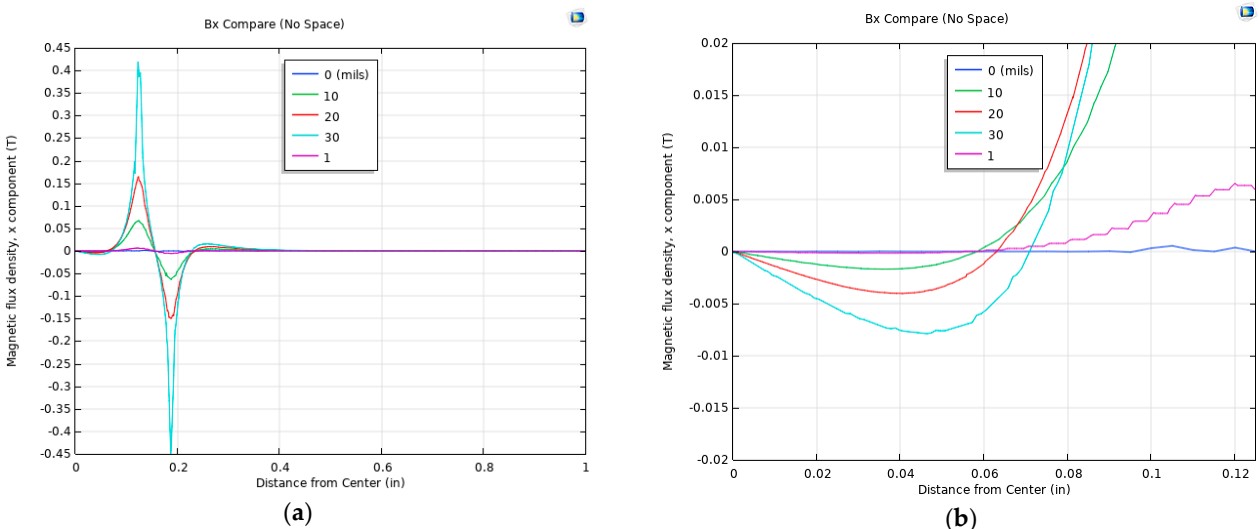

**Figure 14.** $B_x(x)$ at varying longitudinal offset (z-axis, in mils): (**a**) wide x range; (**b**) small x range. Simulation uses no inner to outer pair spacing.

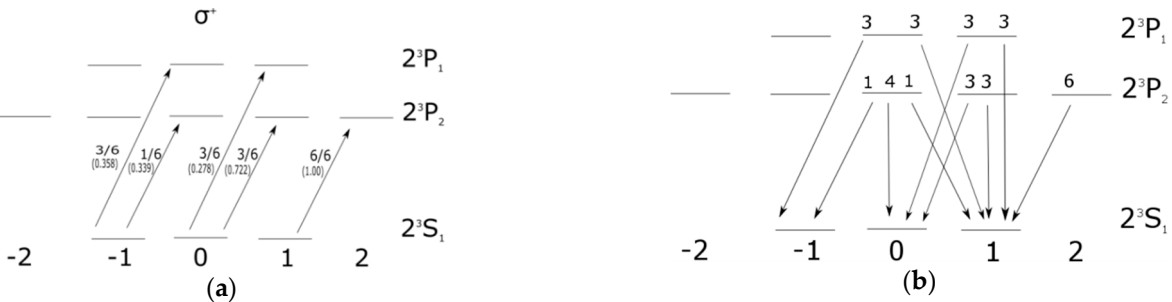

**Figure 15.** $\sigma^+$ $^4$He excitation and relaxation electric dipole matrix elements (0 G, normalized branching ratios labeled on arcs): (**a**) excitation, values in parentheses for 810 G; (**b**) relaxation.

Approximate exponential decline in signal channel population with laser power is determined by the atom-photon cross-section [16] as assumed in Equation (15). Transitions are occurring for many of the available transitions between magnetic field-dependent energy levels that are reachable by the ~2 GHz bandwidth and ~2 mm waist laser, as shown in Figure 15. Reaching 99.3% clearing is a validation of B-field 1/10 collimation nonetheless. Otherwise, significant parallel polarization ($\Delta m_j = 0$) transitions would wash out clearing by returning atoms to the signal channel even at low power. Transit time constraint is known to be power-independent from previous work [4].

Figure 16b adds detail to the high power or infinite power asymptote survival ratio. A pure $\Delta m_j = +1$ model (Model) predicts 0.01% survival while we measure 0.7%. A modeled 0.07 rad misalignment of $\vec{k}$ and $\vec{B}$ (Model 0.07) matches the measured survival asymptote casting further interest on the gap fringe field mentioned earlier. Modeling results are the same for 0 and 810 G within experimental uncertainty. In addition, low to intermediate power survival ratio fluctuations in (a) are now clearly visible in (b). The Yb-doped fiber preparation laser is known to hop between 3 MHz separated modes (50 m cavity) therefore stimulating the 1.6 MHz FWHM transitions in fluctuating fashion.

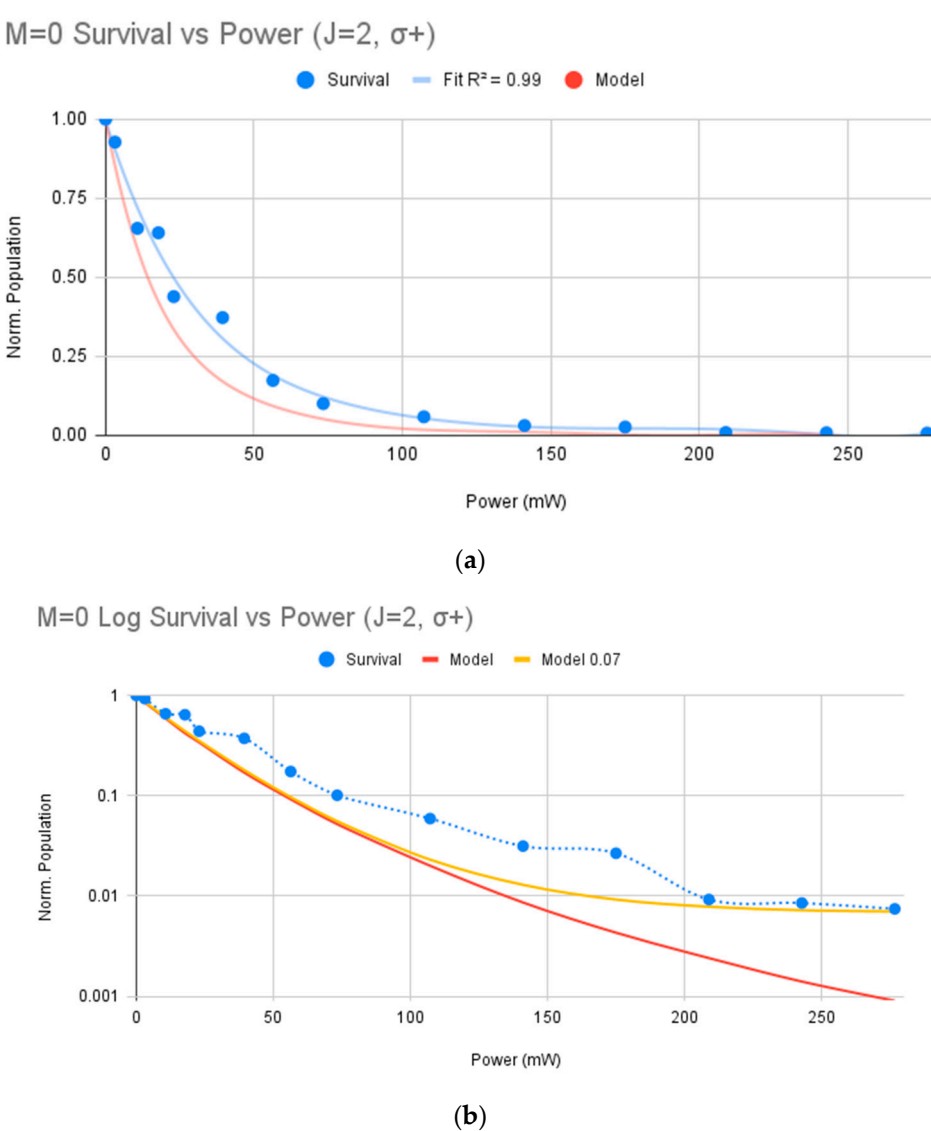

**Figure 16.** Preparation laser survival ratio (1 μm, σ⁺/left) vs. power: (**a**) linear plot and base model; (**b**) log plot and 0.07 rad misalignment model.

Experimentally, zero/near-zero B-field transit between source and detector causes helium beam depolarization as Zeeman energy level separations disappear/narrow. Interaction region Helmholtz coils are used as a precaution against such B-field zeros.

## 6. Conclusions and Further Work

This miniature magnet has been shown to produce an 800 G, well-collimated axial B-field (z-axis) to better than 1/10 radian, as verified by simulation, tabletop measurements, and atomic experiment (Figures 11 and 16). The fringe field falls as an octupole $(1/r^5)$, reaching a small magnitude and low gradient residual dipole far field (arising from imperfect dipole cancelations) in less than 3″ (7.6 cm), as determined by both simulation and experiment (Figure 13). Any space-constrained application needing a non-shielded, localized, and well-collimated magnetic field source can benefit from this device or similar design.

Some improvements possible from this effort are:

1.  *Balancing magnet pairs* will improve far-field cancelation and will also improve central field collimation;

2.  *Further pumping modelling and measurements* will improve understanding and possibly increase signal clearing beyond 99.3%; in particular, study reducing the laser pumping near the atomic beam exit-side of the magnets where Bx is significant; and
3.  A *30 GHz **broadband laser*** will allow experimentation on 3He and 4He simultaneously.

**Author Contributions:** G.C. wrote the original draft. All authors partook in conceptualization, software, formal analysis, investigation, and review and editing for the manuscript. All authors have read and agreed to the published version of the manuscript.

**Funding:** This research was funded by National Science Foundation (USA), grant number 1404498. The APC was funded by the same.

**Institutional Review Board Statement:** Not applicable.

**Informed Consent Statement:** Not applicable.

**Data Availability Statement:** Not applicable.

**Conflicts of Interest:** The authors declare no conflict of interest.

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
