# Peer review of "A Miniature Permanent Magnet Assembly with Localized and Uniform Field with an Application to Optical Pumping of Helium"

_applsci, doi:10.3390/app11198886_

Round 1
Reviewer 1 Report
This manuscript is devoted to development compact and uniform magnetic field source for using in atomic laser spectroscopy experiment (e.g. high precision 3,4He laser spectroscopy experiment). The simulations of characteristics of this magnetic field source, choice of its parameters (all dimensions) and experimental investigations were carried out for designing, building, and testing this device.
Authors improved the manuscript, but some points concerned the figures mainly are required to correct.
- Some captions in the Figures are very small in new variant of manuscript (e.g., all captions of states in new Figure 1). If authors draw the axes (Figures 2), it is necessary to mark these axes (in previous version the X,Y,Z-axes were marked). The numbers in captions of axes in right upper part of Figure 3 are very small. There is not negative part of X-axis in low part of Figure 3.All captions in all of three parts of Figure5 (a-c) are greater than previous version, but small for reading. All captions of axes in figure 7 (a,b,c) are very small for reading.
- It is necessary to control the formula upper 17th line of page4: inside the magnetic material B=μH (as in formula upper 9th line of page 4), but not μ0 (as in formula upper 17th line of page4).
- The formula upper 5th line at page 6 has not any sign (e.g.,“equal to”) before lim.

Reviewer 2 Report
Optical pumping is a well-established tool in many atomic physics experiments. On the way to higher precision and accuracy, many incremental improvements are needed to reach minor and major goals. In such a context, the manuscript is most welcome to Atoms.
Page 2, line 24: "Currie point" might be an Indian food truck; could it be that "Curie" (as in "Madame Curie") is closer to the point (or Curie temperature)?
Page 3, line 1, something is missing in this sentence. The style of English is slipping as well; please reconsider what you want say and say it more clearly and in a less hasty way.
Laboratory slang may be good enough among close colleagues, but you want to interest readers from farther away, too.
Page 4, line 2: "Areas beyond 3” from the Gap 2 center comprise the far field while the Gap is labelled." Translate into understandable English with a notable sense of physics, please.
There are MANY more sentences of non-standard English grammar - why are you confusing the basics of language?
Line 15, "Closeness of simulated and measured values described later indicates that this effect is a small contribution."
Reconsider the logic and meaning of the text.
Page 5, line 15, "Biot-Savart", not "Biöt Savart"
Page 8, figure 11: There are unexplained small numbers "2" at the de-excitation arrows.
The text above the figure is poorly phrased.
Below the figure, the units of measurement are italicized - why? Variables ought to be in Italics (as they are), not such units.
Page 12, figure 16: more of the unexplained small numbers.
In conclusion, the project report is publishable in Atoms, but the present text is not. Have somebody with some better feeling for spoken and written English advise you on the wording of the text. The present version is atrocious - from a (largely) English speaking university.
Reviewer 3 Report
Magnetic field measurement methods have not been explained enough. A curve showing simulation results vs measured field is more than welcome.
Round 2
Reviewer 3 Report
Much improved since the first version concerning my comments.
This manuscript is a resubmission of an earlier submission. The following is a list of the peer review reports and author responses from that submission.
Round 1
Reviewer 1 Report
This manuscript is devoted to development of very compact and strongly collimated magnet for using in atomic laser spectroscopy experiment (e.g. high precision 3,4He laser spectroscopy experiment). The simulations and experimental investigations were carried out for designing, building, and testing this device.
There are some points to correct or to make the information more clear:
- First of all it should be noted that if authors said about far field and made the simulations, it is necessary to mark, what does far field mean for this magnet system.
- The text is careless written in some places:
- There are, e.g., disappearance of the references in text (“Error! Reference source not found..”) – lines with numbers of 34, 53, 58, 66, 85,104, 119, 146, 148, 162-163, 173, 174, 181, 189, 203, 204, 213, 214, 221, 227. It is possible that it is references to the Figures, because there is not any mention about first Figure 1, Figure 2, Figure 5, Figure 7, Figure 8 etc. in the text. It is not clear at the reading.
- There are not the names of physical values in the text (in lines with numbers of 30) or after the formulas (in lines with numbers 39, between the 72nd-80th lines, between 109th-114th lines, between 126th -127th lines, between 158-159 ). It is necessary for understanding the text. Besides sometimes N is number of coils in solenoid, sometimes N is population density in atomic state.
- All abbreviations must be explained at the first mention in the text. But there are some abbreviations without this explanation (e.g. OAM (10th line); OD and ID (13th line); COMSOL (98th line), EMS (187th line), LDD (237th line)). Besides, the abbreviation in abstract (OAM) is nowhere further used.
- Sometimes there are not word spaces and punctuation marks (or excess ones)(e.g., “loops(Multiple” (105th line)), “plots:.(a)” (132nd line), “surfaces(Colloidal” (143rd line)
- One sentence is presented twice (“Exploration Finite Element Analysis (FEA) simulations shown in Figure 1 examine different configurations of similar annular magnets” 88-89th lines and 93-94th lines).
- Two Figures have number 1 (29th and 97th lines).
- What is the “ppg 225” in 126th line? Is it reference to the p.225 in [7]?
- The goal of work is usually presented in introduction. But it is not presented here.
- It is not convenient, that there are two units for B in the figures (sometimes it is Tesla, sometimes it is Gauss).
- There are small indexes in the caption in the Figure 2 (upper part). There are not units in the captions of axes in Figure 2 (lower part) and the numbers in captions of axes are much broadened (the picture resolution is very low); besides they are presented in positive part of X-axis only. There are only one axis without any caption and units in the right upper part (and there isn’t vertical axis). It isn’t clear from the text that the design developed by authors is presented in Figure 2 or it is from the other reference.
- There are very small captions and x,y,z,-axes in the Figure 3.
- Notes concerning with second Figure 1 at 97th First of all, it must be Figure 4. All captions in all of three Figures (a-c) are very small and unreadable. What is the value (and its units) presented at Y-axis? What is abbreviation of AR in Figure cutline? All abbreviations must be explained at the first mention in the text, but it is possible that this abbreviation is explained twice at 203rd and 207th lines (more than 100 lines below). There is not any mention about (b) part of this Figure in the text.
- In formula between 126th-127th lines Bz(theta1=0, theta2=0)=mu0*N*I exactly. It must be mark of “=”, but not arrow.
- Notes concerning with Figure 6 at 132nd All captions in all of four parts of Figure 6 (a-d) (all values and numbers in all axes) are very small and unreadable. It is necessary to increase all captions and letters and also resolution of figures. The formula “i.e. actual Gap (1/16”) + 2 * coil offset (1/32”) – Attracting Pair (Red), Repelling Pair (Blue), Sum (Green)” in the cutline of Figure 6 (135th line) is not clear, it is necessary to explain it. Where are the color scale and its explanation in the (c) of Figure 6?
- The legends of arrows in the Figure 7 are very small.
- In spite of the fact that dimensions of captions in the Figure 8 are greater than in some previous Figures, they are necessary to be increased.
- Notes concerning with Figure 9 at 165th All captions in all of three parts of Figure (a-c) are very small. Besides it is necessary to recognize clearly the directions of transitions (they are almost invisible).
- It is not clear, what Figure is explained in the 173rd -183rd
- Notes concerning with Figure 10 at 191st The legends and captions of axes must be increased.
- Notes concerning with Table 1 (207th line). First figure (AP(1/16”) and Axial (z-axis) Simulation) has three colors in legend (but legend is cut at the beginning and at the end), but there is only blue curve in this figure. What does 0.0625 mean? What does 0.125 in low Figure (for AP(2/16”)) mean? In the two figures for AP(1/16” and 2/16”) B is presented in Tesla, but in two next figures (Transverse (x-axis) Empirical Measurement) B is presented in G. It isn’t convenient for comparison. There are also small caption of axes in these (Transverse (x-axis) Empirical Measurement) both figures. What means “knee”? If it is sharp turning of curve, than the values presented are not corresponded to values in these curves.
- Notes concerning with Table 2 (208th line). There are also small captions of axes in first figure (AR(1/32”) Axial (z-axis) Simulation). What does 0.0625 here and below mean? There are also small caption of axes in next both (Transverse (x-axis) Empirical Measurements) figures. B is presented in Tesla and Gauss in neighbor figures again.
- There are very small captions of axes and legends in the Figure 12.
- 242nd line: 3” is equal to 7.5 cm (more exactly 7.62 cm), but not 7.5 mm as written.
- There is no any comparison of results presented with experimental or simulated data of other authors.
This manuscript describes with details the results of simulations at developing the magnet assembly and experimental control of this magnet assembly. In general the text is unsufficiently clearly written because all points cited above make difficulties for reading.
The manuscript can be published after minor revisions.
Reviewer 2 Report
The manuscript, `Miniature Annular Permanent Magnet Assembly with Fast Far Field Decay and Majority Collimated Central Field' reports development of a compact magnet system with four permanent magnets for He-3,4 laser spectroscopy. The manuscript concludes that the preparation efficiency reaches 96% with this magnet system, and that 99.99% efficiency can be reachable.
However, it is hard for me to confirm the conclusion, and I cannot recommend the publication of this manuscript for the following reasons.
a) Almost all figures are tiny, poor resolutions, and very hard to read even the labels in the figures. The references of the figures in the article show compilation errors (Error! Reference source not found.). In addition, there are two `Figure 1' in p. 1 and p.4.
b) There is little information on the permanent magnets used. For example, no surface field strengths, uniformities, individualities are shown.
c) The magnetic field is calculated by many methods (FEM, Biot-Savart, COMSOL, EMS,...) and shown in many figures, but there is little validation for the calculated results. The comparison with the real magnetic field is only shown in Fig. 10(b), and there are no comments for the discrepancy.
d) In Sec.4, the authors optimize the magnet system by changing the AP/AR gaps. But the definitions of the AP/AR gaps are not explained. In addition, I can't figure out where are the 8G knees in the figures in Table 1.
e) There are no quantitative descriptions of the relation between the magnetic field and the preparation efficiency. Since the preparation efficiency depends not only on the magnetic field, but also on the properties of the laser and the atomic beam, it is unclear that the magnetic field satisfies the requirement only from the measured 96% efficiency. It is also unclear that the increasement of the laser power can raise the efficiency until 99.99%.
Reviewer 3 Report
I have read the manuscript "Miniature Annular Permanent Magnet Assembly with Fast Far Field Decay and Majority Collimated Central Field". Major editing and integration is necessary before any judgement can be made. In particular:
- Most of the references to figures are errors
- The introduction does not contain enough information and must be greatly expanded introducing clearly the apparatus
- Many paragraphs contain unrelated phrases making very difficult to follow any logic.
- Many symbols are not explained
- All the figures have axes labels which are unreadable
- Acronyms without a definition including the abstract
The paper in its current form is unreadable and incomprehensible. Let me ask the authors to try having an external physicist read their paper.
Round 2
Reviewer 2 Report
I think this revised version is not enough answer for my comments.
Some figures are still very poor qualities, no 8G knees are shown in the figures, no detailed laser/atomic beam properties used are described. Especially, no validations are added to the magnetic field calculations. Although the calculations of the magnetic field are one of the main topic of this manuscript, their correctness is still not confirmed by measurements.
I cannot recommend publication of this manuscript.
Reviewer 3 Report
I have read the revised version of the manuscript. Unfortunately the paper is still not in a condition to be published. The paper has been slightly improved but several figures are still unreadable and several paragraphs are still not comprehensible (for example the first paragraph of section 2 just to cite the first). I really have trouble reading the paper.
Line 19 page 2: Be careful, Φ in the equation just above is not the magnetic flux but the magnetic scalar potential.
As I suggested in the first round, the authors should have some external physicist read the paper. I also don't understand if, besides the corresponding author, the other two authors have proof read the paper.
It's too bad because the information in the paper is interesting.